# A Computational Study on the Role of Lubricants under Boundary Lubrication

Walter Holweger [1,2,*], Luigi Bobbio [3,*], Zhuoqiong Mo [4], Jörg Fliege [3], Bernd Goerlach [5] and Barbara Simon [1]

1 Technology Consultant, Sailegärten 2, 72351 Erlaheim, Germany
2 Operational Research Group, School of Mathematical Sciences, University of Southampton, Southampton SO17 1BJ, UK
3 School of Mathematics, Faculty of Social Sciences, University of Southampton, Southampton SO17 1BJ, UK
4 Business School, Faculty of Social Sciences, University of Southampton, Southampton SO17 1BJ, UK
5 ASC Goerlach, Robert-Bosch-Straße 60/1, 72810 Gomaringen, Germany
* Correspondence: walter.holweger@t-online.de (W.H.); l.bobbio@soton.ac.uk (L.B.)

**Abstract:** The knowledge of how lubricants contribute to the operational life of a drive train is unclear until now, despite the fact that plenty of literature is available. A novel concept is presented in order to estimate the wear appearing in bearings addressed to the regime of mixed friction with respect to the composition and the so-called "inner" structure of the lubricant. In doing so, the composition is turned into a set of predictors describing the dipolar and inducible dipolar properties of all components as an activity amongst them and toward the surface. The results show that the activity of the solvated specie apparent, stated as the "inner" structure of the lubricant, is closely related to the surface activity and the expected wear. The technique presented here allows a fast computational procedure such that a given lubricant, once known by its constituents, could be explored with respect to the expected wear. Reducing time-consuming tests is desirable by the fact that new materials are forthcoming as a consequence of regulations and evolving green technology.

**Keywords:** lubrication; bearings; wear

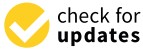



## 1. Introduction

The increase in energy density by size reduction and power throughput combined with life installation of bearings in industrial and automotive drive trains has intensified the research on how lubrication could contribute to these demands. Traditionally, lubrication is thought to separate mating surfaces by the formation of a liquid film, covering the surface asperities. A connection between the film formation capability and the physical properties has been established as the theory of elastohydrodynamic lubrication (EHL) comprising the viscosity and its relation to temperature, pressure and miscellaneous parameters by Dowson–Higginson [1]. However, within the operational mode in the life cycle, film formation may not be constant and lead to thin film formation (TFL), partially uncovering some asperities under mixed friction. TFL has become a field of intense research in the past [2]. While for standard lubrication, the asperities are covered as thin films and very thin films appear, the properties of lubricants with respect to their molecular structure are becoming important. In TFL, the parameters of viscosity are calculated on the base of molecular dynamic (MD) simulation [3]. However, the role of chemistry in TFL is not to be neglected. If the mating surfaces are partially in touch, chemical reactions catalysed by metal and oxygen take place and tribocatalysis takes place as an aspect of catalysis (TC) [4]. The interaction of chemicals with metal surfaces has been studied over the years [5,6]. As standard steel surfaces are usually covered by thin metal oxide layers, their reactivity becomes essential [7]. The role of different surface oxygen species is described in [8]. Tribochemical reactions as a function of contact mechanics are described by Spencer, et al. [9]. Detailed studies comprising the influence of the chemistry of lubrication additives on gear

life are reported by [10], stating a relation of a given chemical structure with the nominal life of drive train components by a structural–property relationship. The appearance of white etching cracks (WEC), mainly addressed to bearings, has brought up similar results [11–13]. A machine learning study was carried out by [14,15] in order to reduce the dimension of parameters and come to data separation. However, bringing the role of lubrication chemistry down to irreducible parameters with respect to the interaction with materials present in a drive is still missing. This search has to be addressed mainly to the boundary and mixed friction regime in the Stribeck curve, whereas the regime of full film lubrication therein is still satisfactorily described by EHL. If, by definition, the boundary regime and the mixed friction regime are governed by TFL, TC also becomes evident, especially the relation between TFL and TC. Both theories have to be merged and therefore the matter of irreducible predictors in lubrication becomes evident. However, the processes involved in the surface–chemistry interaction have to be seen in detail. Chemical reactions that are supposed to be essential in boundary and mixed friction lubrication chemistry do not take place in a simple manner, since lubricants are mixtures covering different functional media, such as the base oil, but also functional additives dissolved in the base oil. If the lubricants come into contact with the surface as assumed in the TFL regime, it is important how the different players behave and how they are attracted or rejected by the surface in the first step. Therefore, it becomes essential to describe how a surface selects plenty of different molecules being mutually dissolved. Modern computation takes care of these interactions and the transient processes appearing by applying a molecular dynamics simulation to multicomponent systems [16–18]. However, despite these concepts lead to an insight from an atomic to a mesoscopic scale, the computational effort is high and rarely applicable to very complex mixtures in a reasonable time. Starting from molecular dynamics, Density Functional Theory (DFT) and Parametric Method 3 (PM3) simulations [19–23], the current study aimed to determine whether the current simulation techniques as cited in [16–18] could be facilitated to gain fast computing of complex mixtures with plenty of functional additives exposed to a steel surface with varying activity.

## 2. Materials and Methods

### 2.1. Theoretical Approach

This approach calculates the basic properties by the use of the density functional theory (DFT) and PM3 calculation, such as the dipolar and induced dipolar activity of a molecule, seen to be essential for the interactions within the lubricant and with a steel surface. Combining these principal aspects with Boltzmann statistics and phase equilibrium thermodynamics, we developed a simple model suitable for complex mixtures and their tribology. A solute representation of a lubricant has to address all the interactions among its constituents. The interaction takes place due to polar attraction (ions, dipoles) and induced dipolar attraction (polarisability). Moreover, all the solute specie reaching the surface of the contact body will interact with it. Solute and surface attractions may lead to different conditions. If B is a base oil and A is an additive, the solute structure may be described as:

1. Solvate BxAy, with x >> y, meaning that plenty of base oil molecules cover the additive;
2. Clustering Bx [Ay]g, with x ~ y, meaning that B dissolves additives clustering together. In extreme cases, micelles are formed with a shell of base oil encapsulating self-aggregates of the additives, expressed as degenerated by a degree of $g$;
3. Clusters of the base oil [B]$g$ temporarily created random with a degree of $g$;
4. Clusters of the additive [A]$g$ temporarily created random with a degree of $g$;
5. Adsorption at surface sites S:

    5.1. Solvates from (1) interact: {[BxAy]$g$}-S either single with a degeneration $g = 1$;
    5.2. Clusters ($g > 1$) from (1) {[BxAy]$g$}-S;
    5.3. Base oil directly [B]$g$-S with $g = 1$;
    5.4. Additives directly [A]$g$-S with $g = 1$;
    5.5. Base oil oil clusters [B]$g$-S with $g > 1$;
    5.6. Additives clusters [A]$g$-S with $g > 1$.

The reader may also refer to Supplementary Materials, where we provide detailed explanation and examples of the symbols and situations used above. In what follows, we assume the interaction of molecules to take place via their permanent dipoles and their inducible dipoles as polarisability. Any individual molecule may then be described non-dimensionally as a dipolar and polarisable parameter.

$$a(\mu_i) = \frac{\mu_i}{\sum_{i=0}^{i=m} \mu_i} \tag{1}$$

Dipolar activity of the *i-th* molecule with $\mu_i$ dipole moment of the *i-th* molecule.

$$a(\alpha_i) = \frac{\alpha_i}{\sum_{i=0}^{i=m} \alpha_i} \tag{2}$$

Induced Dipolar activity of the *i-th* molecule with $\alpha_i$ induced dipole moment of the *i-th* molecule.

Combining these activities with the Boltzmann equation, it is possible to write

$$a_i = Q g_i e^{-\frac{e_i}{kT}} \tag{3}$$

with $a_i$ as the generalized activity, either to be dipolar or induced dipolar; $k$ as the Boltzmann constant; $T$ as the absolute temperature, indexing the *i-th* component; and $g_i \geq 1$ as degeneration factor and $Q$ a normalisation constant, implying $0 < a_i < 1$.

Solving this equation leads to

$$e_i = -kT * \ln\left(\frac{a_i}{Q g_i}\right) \tag{4}$$

Similar for a specie $j$, we formulate

$$e_j = -kT * \ln\left(\frac{a_j}{Q g_j}\right) \tag{5}$$

where $e_i$, $e_j$ are then the energies of the *i-th* and *j-th* specie, respectively, while $a_i$, $a_j$ are the dipolar or induced dipolar activities of the respective species. As a temperature independent, non-dimensional parameter we define

$$D_{i,j} := \frac{e_i}{e_j} = \frac{\ln\left(\frac{a_i}{Q g_i}\right)}{\ln\left(\frac{a_j}{Q g_j}\right)} \tag{6}$$

Assuming the terms $kT \ln\left(\frac{a_i}{Q g_i}\right)$ to be identical to the chemical potentials $p_i$ in a solute of specie $i$, then an equilibrium is defined by the relation $p_i = p_j = \ldots$, or equivalently by $D_{i,j} = 1 \; \forall i, j$. Situations where $D_{i,j} \neq 1$ mean that there is an imbalance in the system. If an imbalance is present in the system, it is reasonable to focus on couples $i, j$, such that $0 < D_{i,j} < 1$. In fact, if $D_{i,j} > 1$, we can then consider $D_{j,i}$. Moreover, it is possible to empirically observe that when $D_{i,j} > 1$, a tendency of the clusters' formation appears. In other words, more species tend to transit to the same energy level $e_j$, hence causing degeneration, i.e., $g_j > 1$. When the clusters are formed, keeping all the other variables the same, the minimum number of clusters $g_j \in \mathbb{N}$, $g_j > 1$, such that $D_{i,j} \leq 1$ is computed and the corresponding value of $D_{i,j}$ is considered. Numerically, $D_{i,j}$ describes the acitivity $a_i$ within the base of $a_j$, hence the strength of its interaction.

For computing the individual dipolar and induced dipolar activities, we used the semi-empirical parametric model calculation 3 (PM3) (Hyperchem$^{®®}$ 8.0), being well aligned with density functional theory (DFT) calculations [16–20]. The base of each calculation starts with the chemical structure of the molecule. After drawing the raw structure, the

molecule is aligned with respect to the optimal bond angles and bond lengths. This state without energy relaxation is declared as $\overline{a_i}$: the upper state of the activity. The molecule is then relaxed by geometry optimization in molecular mechanics (MM+). The ground state is defined by a root mean square (RMS) parameter 0.01. After relaxation, the energy is recalculated by PM3 and defined as $\underline{a_i}$: the energy state of the MM+ relaxed molecule. The definition of any *i-th* molecule $m_i$ is then given as

$$\underline{a_i} < m_i < \overline{a_i} \tag{7}$$

with $0 < \underline{a_i} < 1$, $0 < \overline{a_i} < 1$ and, implicitly, $\underline{a_i} < \overline{a_i}$.

Figure 1 gives a flow chart for the calculations. The skeleton structure is initially modelled by adding hydrogen and the energy calculation of the non-relaxed structure is set as $\overline{e_i}$ for the *i-th* specie and the same for the *j-th* specie as $\overline{e_j}$. After relaxing within molecular mechanics (MM+) to an RMS value $< 0.01$, the energy is calculated by MM+ as $\underline{e_i}$ for the *i-th* specie and $\underline{e_j}$ for the *j-th* specie. Normalizing the energies for the *i-th*, similar for any other specie, we obtain the dimensionless parameter.

$$m_i(up) = \overline{m_i} = \frac{\overline{e_i}}{\overline{e_i} + \underline{e_i}}$$
$$m_i(low) = \underline{m_i} = \frac{\underline{e_i}}{\overline{e_i} + \underline{e_i}} \tag{8}$$

The *i-th* molecule $m_i$ (similar to all other index molecules) is considered to exist in between the ranges of low and up.

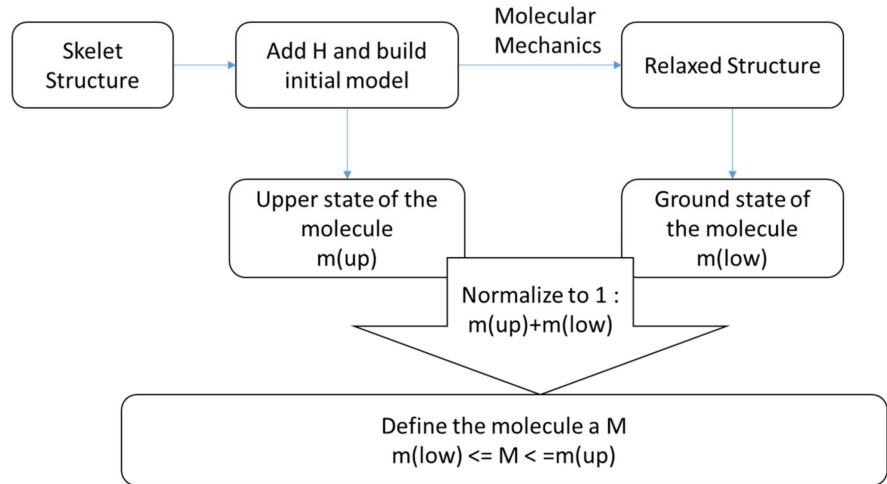

**Figure 1.** Flow chart.

The surface–molecule interaction is calculated similarly, assuming that the surface is defined by its activity *s*. Defining *s* assumes a thin oxide, dipolar layer is laying on it, as confirmed by secondary neutral mass spectrometry (SNMS). This layer may act as a dipolar partner like any other competitive molecule. Similar to molecules, the surface is defined by its low and up activity, $\underline{s}$, and $\overline{s}$.

$$S : \underline{s} < s < \overline{s} \tag{9}$$

As this parameter is not explicitly calculated by a method, it is treated as a free parameter. The general interaction term $D_{i,j} = \ln\left(ci * \frac{a_i}{Qg_i}\right) / \ln\left(cj * \frac{a_j}{Qg_j}\right)$ is now computed over the entire range

$$\underline{a_i} < a_i < \overline{a_i} \text{ and } \underline{a_j} < a_j < \overline{a_j} \tag{10}$$

taking two partners into the interaction for simplicity and $c_i$, $c_j$ to be the normalized concentrations ($0 < c_i$, $c_j < 1$). The degeneration terms $g_i$, $g_j$ are assumed to be 1 (no degeneration) as long as

$$0 < D_{i,j} < 1 \qquad (11)$$

If a system of interactions degenerates, it is called a cluster as an aggregation of molecules in the same state. Furthermore, the interaction is only then computed if the molecular states are within a selected distance $t$.

$$t = \left| m_i - m_j \right| \qquad (12)$$

As plenty of lubricants are found by their brand names, chemical abstract service numbers (CAS) or any other standard, it is easy to review their structures via a web search. A different method for determining the structures is given by analytical procedures. As a standard, the lubricants are analysed by X-ray fluorescence (XRF) with the output of the elements with atomic masses $> 10$. The elements that are recorded are allowing the recalculation of the structure of the component based on the experience that the chemicals reflect a given structure and the additives normally do not exceed 3% per weight in a lubricant. Therefore, considering the aspects of stoichiometry and the restriction of the percentage of a simple XRF analysis could lead to the molecular structure of the additive by computational best hit search. For the current research, we took a set of structurally well-defined additives defined by their use as extreme pressure (EP) additives. The other set was given by their brand name and their assumed composition was reconstructed with respect to their structure by the use of their spectral and other data.

### 2.2. Experimental Approach—FE8 Test Rig

For estimating the oils' properties, a standard FE8 test rig (DIN 51 819) was used, see Figures 2 and 3, and the detailed test condition could be described as follows:

**Type of bearing**

81212 Cylindrical roller bearing (CRB)
Axial Load: 80 kN
Temperature (held constant): 80 °C
7.5 rpm
Contact pressure: 1890 MPa, 15 Rollers

**Cage**

Brass
Housing washer
Outer diameter: 95 mm
Bore diameter: 62 mm

**Shaft washer**

Outer diameter: 95 mm
Bore diameter
Mean diameter of the bearing: 78 mm

**Bearing material**

SAE: 52100
Martensitic heat treatment
Hardness: $800 \pm 20$ HV
Residual stress: +10 MPa (tension)
Retained austenite: 10–12%
Roughness Rq (washer) 0.02–0.04

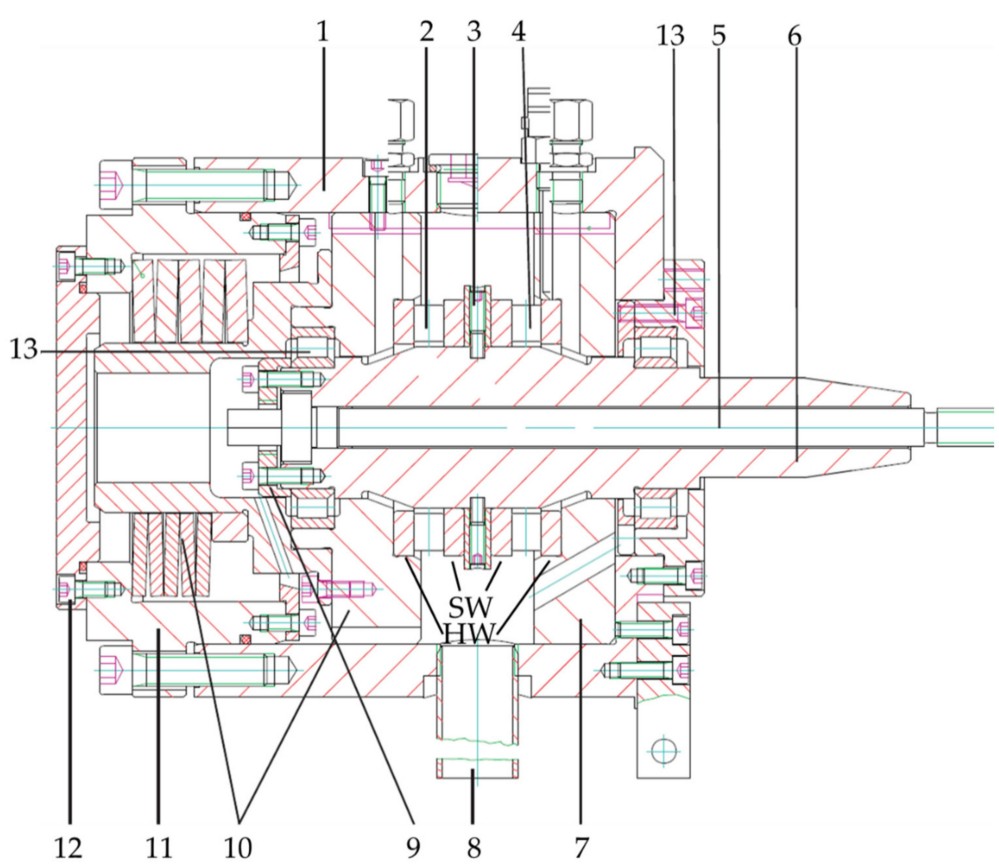

**Figure 2.** Test head with axial cylindrical roller bearings. 1. Housing; 2. Test bearing 2 (test head sided bearing); 3. Spacer; 4. Test bearing 1 (motor sided bearing); 5. Shaft; 6. Clamping bolt; 7. Bearing seat; 8. Drainpipe; 9. Cap; 10. Bearing support with screwed-on pilot pin; 11. Lid cup of spring package; 12. Lid; 13. Auxiliary bearing. Both test bearings consist of a stationary housing (HW) and a rotating shaft washer (SW). Figure adapted from Ref. [24] Copyright 2017, Schaeffler Technologies AG & Co.

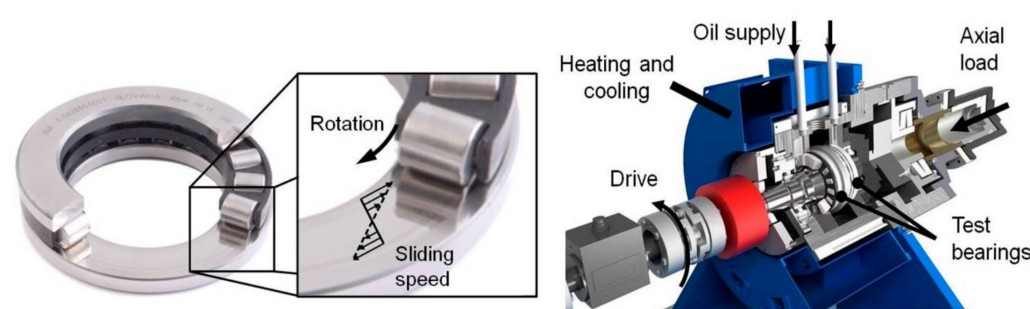

**Figure 3.** FE8 test rig [taken from Gachot, et al. [13].

### 2.3. The Lubricants

For the tests presented, a series of extreme pressure (EP) additives were chosen. As the current study aimed at assessing biodegradable oils, we considered a trimethylolpropane as the base oil for the testing additives. In Figure 4, the chemical structures of the base oil, as well as the EP additive are depicted.

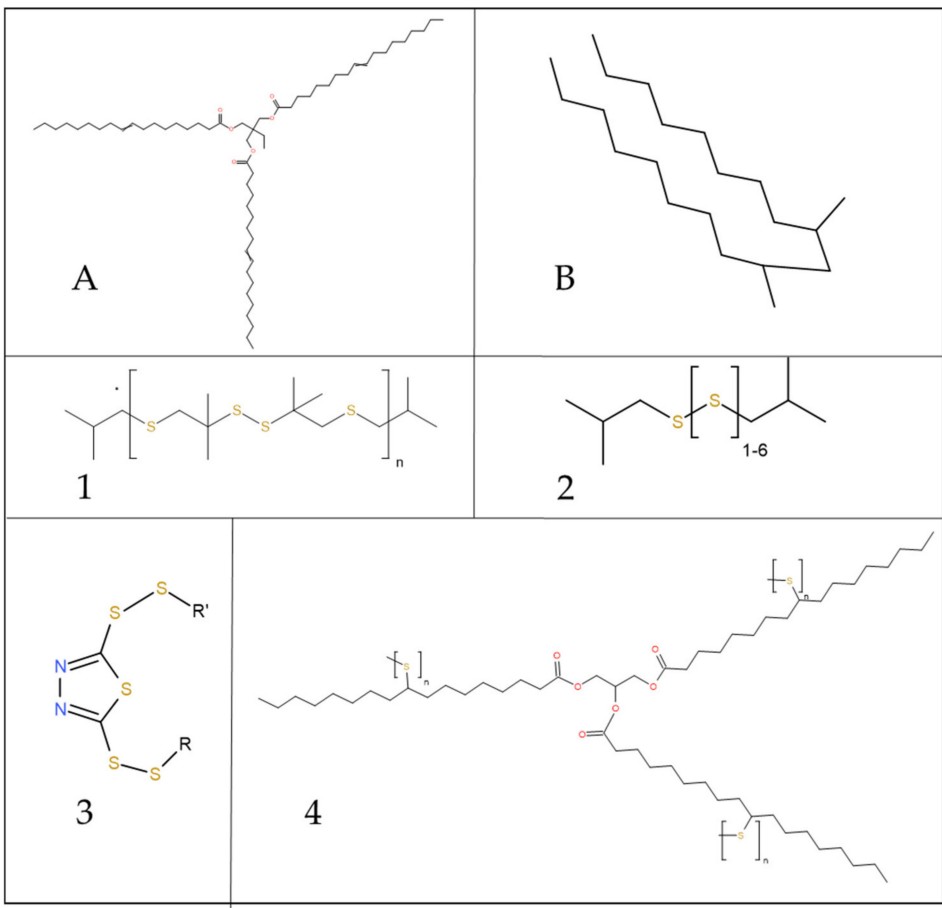

**Figure 4.** Trimethylolepropane trioleate (TMP) (**A**) as a base oil and in two cases, mixtures of TMP and Poly-a-Olefine (PAO) (**B**) for the chosen EP additives 1–4.

The base oil above and the additives were then aggregated in different ways, leading to a set of training lubricants. In Table 1, we show the considered compositions of the lubricants.

**Table 1.** First training set.

| Lubricant Tag | PAO [%]w | TMP [%]w | Additive 1 [%]w | Additive 2 [%]w | Additive 3 [%]w | Additive 4 [%]w |
|:---:|:---:|:---:|:---:|:---:|:---:|:---:|
| L1 | | 97 | 3 | | | |
| L2 | 87.3 | 9.7 | 3 | | | |
| L3 | | 97 | | 3 | | |
| L4 | 87.3 | 9.7 | | 3 | | |
| L5 | | 97 | | | 3 | |
| L6 | 87.3 | 9.7 | | | 3 | |
| L7 | | 97 | | | | 3 |

In addition to the lubricants above, we considered a second set (lubricants L8–L19), where the lubricants' properties were analysed according to their spectral data and the data available within their data sheets.

## 3. Results

*FE8 Test Runs*

Table 2 shows the test rig wear results for the training oils and for the commercial oils with respect to the housing side washer, as being representative for the wear of the components (cage, roller, washer at the motor-side). The authors are aware of the fact that

in mechanical operation, the individual parts (e.g., roller, cage) may interact differently and the results may differ from what is explored here. This will be part of a subsequent paper. Due to the many of overrollings, it is reasonable to look for the wear of the rollers. However, the experiments show that if the rollers are worn, the other parts (e.g., washers and cage) are similarly worn. Therefore, ensuring the usage of data compliant for publishing as well as cases where all the components (cage, roller, washer at the motor-side) are in accordance, we decided to examine the wear of the housing washer for the current study, and we consider it as a good representative of the wear components.

**Table 2.** Wear test results from training oils and commercial brands.

|  | Oil Code | Wear (mg) Housing Washer |
|---|---|---|
| **Training Set** | L1 | 111.5 |
| | L2 | 212.5 |
| | L3 | 170.75 |
| | L4 | 5.75 |
| | L5 | 174.5 |
| | L6 | 153.75 |
| | L7 | 0 |
| **Commercial Oils** | L8 | 0 |
| | L9 | 0 |
| | L10 | 3.1 |
| | L11 | 3.1 |
| | L12 | 3.1 |
| | L13 | 3.1 |
| | L14 | 2 |
| | L15 | 2 |
| | L16 | 5 |
| | L18 | 5 |
| | L19 | 31.75 |

On the other hand, dipole moments and polarisabilities for the base oils, TMP and PAO, as well as for the chosen additives are calculated on the basis of their structure (see composition) by the Hyperchem®® PM3 method in the absence of any external electrical field. Consequently, it is then possible to construct the lubricants predictors P1–P12, resulting from the relative quantities $D_{i,j}$ and shown in Table 3.

The $D_{i,j}$ factors stemming from the dipolar and induced dipolar activity as the assumed "inner" structure of the lubricant, including the degeneration state $g$ (see Equation (6)), are calculated for all the components and their permutations. The dipole moment of iron oxide is assumed to be related as the difference of the electronegativity of iron and oxygen and, as stated, to be present at the near surface as a non-stoichiometric mixture of FeO. More precisely, $Fe_3O_4$ was assumed as a base for the calculation. The study was carried out for a reference surface activity set to 1 as a base. Table 4 shows the results from the training oils and commercial brands (the whole training set), expressed as a percentage of the specie leading to the calculated interaction value $D_{i,j}$, named as predictors P1–P12 and separated, as described above, into dipolar and induced dipolar (polarisability) activity, both as non-clusters and as a clustered specie. In addition, the correlation between each predictor and the FE8 test runs was computed and, as can be observed, a high correlation is found for the predictors P1 and P9 with respect to an assumed surface activity 1 presented

here. We can notice here that it is possible to consider the relations with respect to different surface activities. However, these are part of a current study and are not yet presented.

**Table 3.** Lubricant Predictors.

| Predictor ID | Description |
|---|---|
| P1 | Dipole components of the base oil at the surface |
| P2 | Polarisability (inducible dipole) of the base oil at the surface |
| P3 | Dipole components Base oil and additives at the surface |
| P4 | Polarisability (inducible dipole) of components at the surface base oil and additives |
| P5 | Dipole additives at the surface |
| P6 | Polarisability (inducible dipole) of additives at the surface |
| P7 | Cluster dipole of the base oil at the surface |
| P8 | Cluster polarisability (inducible dipole) of the base oil at the surface |
| P9 | Cluster dipole components of base oil and additives at the surface |
| P10 | Cluster polarisability (inducible dipole) of from base oil and additives at the surface |
| P11 | Cluster dipole additives at the surface |
| P12 | Cluster polarisability (inducible dipole) of additives at the surface |

**Table 4.** Predictors P for the lubricants L1–19 and their linear correlation with the wear of the housing washer.

| Oil | Wear (mg) | P1 | P2 | P3 | P4 | P5 | P6 | P7 | P8 | P9 | P10 | P11 | P12 |
|---|---|---|---|---|---|---|---|---|---|---|---|---|---|
| L1 | 111.5 | 6.16 | 6.08 | 0 | 0 | 0 | 0 | 0 | 0 | 7.65 | 7.61 | 8.81 | 8.86 |
| L2 | 212.5 | 6.29 | 6.17 | 0 | 0 | 0 | 0 | 0 | 0 | 8.8 | 8.74 | 3.97 | 4.01 |
| L3 | 170.75 | 6.64 | 6.5 | 0 | 0 | 0 | 0 | 0 | 0 | 8.92 | 8.85 | 3.81 | 3.85 |
| L4 | 5.75 | 0 | 4.86 | 0 | 0 | 0 | 0 | 7.8 | 0 | 4.22 | 7.29 | 7.92 | 3.66 |
| L5 | 174.5 | 0 | 4.67 | 0 | 0 | 0 | 0 | 3.18 | 0 | 7.5 | 8.52 | 8.73 | 3.28 |
| L6 | 153.75 | 6.98 | 7.1 | 0 | 0 | 0 | 0 | 0 | 0 | 9 | 9.05 | 3.63 | 3.61 |
| L7 | 0 | 0 | 6.39 | 0 | 0 | 0 | 0 | 3.65 | 0 | 2.73 | 8.62 | 8.15 | 3.45 |
| L8 | 0 | 0 | 5.93 | 0 | 0 | 0 | 0 | 3.39 | 0 | 2.48 | 7.67 | 8.83 | 3.63 |
| L9 | 0 | 0 | 5.15 | 0 | 0 | 0 | 0 | 7.41 | 0 | 3.97 | 7.09 | 8.58 | 3.65 |
| L10 | 3.1 | 0 | 6.14 | 0 | 0 | 0 | 0 | 2.07 | 0 | 2.24 | 11.62 | 4.88 | 6.06 |
| L11 | 3.1 | 0 | 7.85 | 0 | 0 | 0 | 0 | 2.4 | 0 | 2.58 | 4.33 | 5.02 | 5.87 |
| L12 | 3.1 | 0 | 6.18 | 0 | 0 | 0 | 0 | 2.42 | 0 | 1.85 | 11.96 | 5.9 | 3.34 |
| L13 | 3.1 | 0 | 7.85 | 0 | 0 | 0 | 0 | 2.47 | 0 | 1.89 | 4.34 | 5.28 | 6.13 |
| L14 | 2 | 0 | 6.61 | 0 | 0 | 0 | 0 | 3.52 | 0 | 2.67 | 8.79 | 3.34 | 3.92 |
| L15 | 2 | 0 | 5.95 | 0 | 0 | 0 | 0 | 3.98 | 0 | 2.98 | 8.93 | 7.63 | 9.67 |
| L16 | 5 | 0 | 7.07 | 0 | 0 | 0 | 0 | 9.63 | 0 | 4.21 | 7.15 | 4.98 | 3.02 |
| L18 | 5 | 0 | 7.14 | 0 | 0 | 0 | 0 | 3.79 | 0 | 2.36 | 6.91 | 3.66 | 4.34 |
| L19 | 31.75 | 0 | 6.42 | 0 | 0 | 0 | 0 | 3.4 | 0 | 2.58 | 8.53 | 8.64 | 3.39 |
| Correlation | | **0.82** | **−0.17** | | | | | **−0.60** | | **0.94** | **0.17** | **−0.18** | **−0.10** |

While the high correlation for factor P1 is neglected since the values are seemingly not scattering due to the presence of many zeroes, predictor P9 shows a comparable strong separation of the values in low wear (corresponding to low P9 values, blue dots) and high wear (red dots) (corresponding to high P9 values), as we can see from Figure 5.

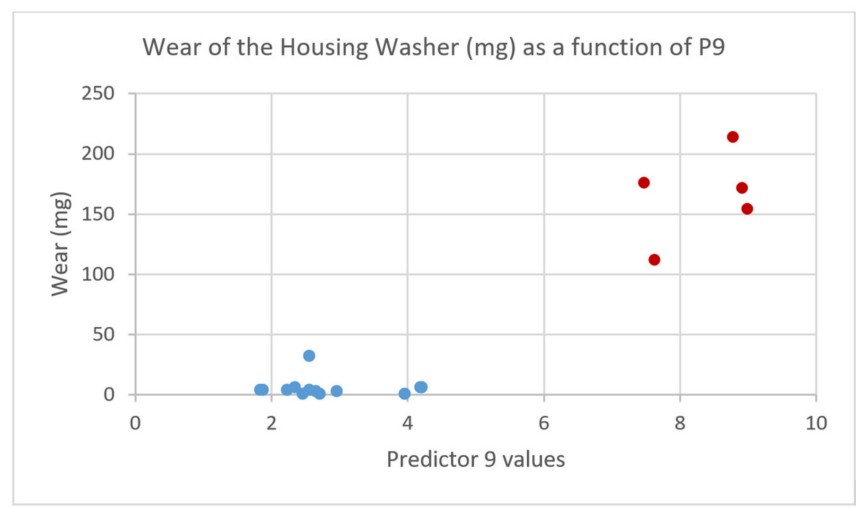

**Figure 5.** Wear (mg) of housing washer as a function of lubricant Predictor P9.

## 4. Discussion

The presented results show a high correlation between the wear apparent in a bearing test rig and the "inner" structure of the lubricant defined by the dipolar properties of the lubricant with respect to an assumed surface activity. It is then possible to understand that the formation of dipolar clusters as a reaction of the "inner" structure of the lubricant onto the surface is unfavourable in preventing wear (see Figure 5). In other words, the fewer clusters, the lower the expected wear. This leads to the conclusions that:

(A)   An oil chemistry could be separated into predictor sets, comprising the (induced) dipolar activities apparent at the surface;

(B)   The (induced) dipolar activity as one predictor of the lubricant corresponds to apparent wear in a bearing test rig;

(C)   The "inner" structure, defined here as the (induced) dipolar activity is a measure of the expected wear in a bearing test rig;

(D)   This strong relation allows the prediction of the expected wear in an FE8 test device as a parametrized bearing application with respect to a given lubricant.

In addition to these statements, it is possible to notice that the formation of dipolar clusters from the lubricant at the surface is clearly related to its activity. As the activity of the surface itself undergoes a change by lubrication, especially in the mixed friction regime, the interaction mode of the lubricant may change as well. In the spotlight of this investigation, it is obvious that estimating solely the viscosity parameters (viscosity–temperature, viscosity–pressure), as assumed in the Dowson–Higginson equation is insufficient in the TFL and TC region as the dipolar activities are changing, hence the mode of the lubricant–surface interaction. With respect to tribology, it is moreover insufficient to argue about the chemical structure and the reaction products in the TC region when predicting lubricants. The assumption of a controlled chemical reaction from the additives at the surface is misleading, as reported already [10]. The predictability of the preventive function of a lubricant is thus related to the region of the Stribeck curve. Under full film operation, viscosity-related parameters are the main predictors, as stated in the Dowson–Higginson equation. The function of a lubricant, however, changes explicitly in the mixed friction and boundary region. With respect to the prediction of the lubricant's function and, subsequently, its contribution in the lifecycle operation of the tribocontact, the traditional method of calculating solely the film thicknesses related to the asperity

profile will not lead to satisfactory results, the same as assuming chemical reactions derived from additives. In the light of this study, in predicting the lifecycle of a tribosystem, one must consider the complete "modus operandi" and the specie involved. The functionality of a lubricant changes with the distance to the surface and causes the surface activity as well. It becomes apparent that processes in the mixed friction and boundary contact of a tribosystem takes place in a sequence of processes, dependent of the distance to the surface. A lubricant possesses in an "inner" structure by solvation of the functional additives and the base oil themselves. This "inner" structure of solvates changes as the lubricant reaches the surface. The surface potential, described here by the relative dipole moment as the attraction principle, interferes now with the solvent structure of the lubricant, setting it under stress. The competition of the solvent structure with the surface first releases the most active specie onto it. This may be either the additives or the base oil components. This situation is highly unbalanced and far from equilibrium. Then, if the impact caused by the surface interaction does not allow for reaching a new equilibrium, the "inner" structure of the lubricant is trying to "repair" it by the creation of new structures and self-aggregations (clusters). The cluster formation makes plenty of lubricant components to behave as a one, hence a "new" big specie. By the self-assembling of functional components as clusters, the original function to bring it toward the surface in order to protect it might be trapped and become lost. In order to disrupt those clusters, the surface activity should be high enough. This brings us now to the point of a further study, where the dependency of the interactions of the lubricants within TFL or TC region on the "inner" structure and the "surface" activity is investigated. Since the surface activity may undergo changes over time, a functionality of the lubricant may also change, either from bad to good or inverse, depending on the conditions. Parameter test rigs often reflect conditions, where a few parameters are tested over time and extreme conditions, while the others are supposed to be kept constant. As the lubricant follows the surface activity, these tests might deviate significantly and even more with respect to the real lifecycle. Within this perspective, the questions around TFL and the TC structural processes within a lubricant cannot be neglected for test rigs, but even more for the lifecycle sustainability.

## 5. Conclusions

In this paper, we analysed and estimated the wear appearing in bearings addressed to the regime of mixed friction with respect to the composition of the lubricant. In order to come to a numerical approach, the lubricants and their additives are associated with "inner" quantities described numerically by simulations. Therefore, pairing FE8 test rig results with a set of predictors stemming from the numerically described lubricants composition as well as their chemical properties stated as activity among all the components and the surface, we have shown that the activity of the solvated specie as an "inner" structure of the lubricant is closely related to the surface activity and the expected wear. In particular, from the given results in this study, of the following conclusions are derived:

(A) A lubricant can approach the surface by transportation across its viscosity, temperature and shearing.

(B) The surface interacts with the most attractive components, defined by the dipolar and induced dipolar interactions, normalized to a "relative" dipole moment as a dimensionless parameter.

(C) The presence of these species is found to be more essential in the initiation of the wear processes in mixed friction and boundary lubrication rather than their assumed chemical reactions.

(D) The nature of the specie, e.g., single or clustered, is thought to be an irreversible predictor for wear.

As a consequence of these statements, we can deduce the importance of coupling an FE8 test rig with analysis of "inner" chemical properties and of conducting future studies to reveal in detail how the "inner" structure of a lubricant is related to the surface activity with respect to parameter test rigs, but even more to the lifecycle operation.

**Supplementary Materials:** The following supporting information can be downloaded at: https://www.mdpi.com/article/10.3390/lubricants11020080/s1. We also furnish the supplementary material *Symbols Explained with Example*, where we describe more in detail the notation used, also providing some examples that help in clarifying the situations considered.

**Author Contributions:** Conceptualization, W.H. and B.G.; Methodology, W.H., B.G. and L.B.; Validation, B.G., B.S. and W.H.; Investigation, B.G.; Resources, B.G., W.H. and B.S.; Data curation, W.H., Z.M., J.F. and L.B.; Writing—original draft preparation, W.H. and L.B.; Writing—review and editing, W.H, B.G. and L.B.; Visualization, J.F.; Supervision. All authors have read and agreed to the published version of the manuscript.

**Funding:** This research received no external funding.

**Data Availability Statement:** Data sharing not applicable.

**Conflicts of Interest:** The authors declare no conflict of interest.

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
