# Peer review of "A Computational Study on the Role of Lubricants under Boundary Lubrication"

_lubricants, doi:10.3390/lubricants11020080_

Round 1

Reviewer 1 Report

The work is focused on a computational study of lubricants role under boundary lubrication. The idea of this paper seems interesting and is presented in a clear way. I have only some minor questions:

What was the standard deviation or scattering of the experimental wear results presented in the paper (only average value is shown). And would the potential large spread of these results affect the correlation with the model?

Where do you see areas where the results could be applied. Do the results have only cognitive or also applicable character?

Reviewer 2 Report

This paper presents a computational study on the role of Lubricants under boundary Lubrication. The topic is interesting, and may be useful for wear prediction. However, the manuscript is not well prepared and written.

Comments:

1.      Equations are not marked correctly. For example line146, equation number is missing. Equations of Lines 165,169,173,177 should be numbered accordingly.

2.      Nomenclature should be provided since they are many symbols.

3.      Experimental details are not described clearly, for example, operating conditions, the material of bearings, wear measurement method

4.      This reviewer has also another comment, the contact behavior of frictional pairs under boundary condition is related to the materials of mating surfaces, surface roughness and lubricant. Did the authors consider the mutual influence of these factors? Discussions are necessary.
